# Flexible Pressure Sensors with a Wide Detection Range Based on Self-Assembled Polystyrene Microspheres

**DOI:** 10.3390/s19235194

**Published:** 2019-11-27

**Authors:** Wufan Chen, Bingwei Wang, Qianbing Zhu, Xin Yan

**Affiliations:** 1College of Information Science and Engineering, State Key Laboratory of Synthetical Automation for Process Industries, Northeastern University, Shenyang 110819, China; 1700731@stu.neu.edu.cn; 2Best on Earth (BOE) Technol Grp Co Ltd., Beijing 100176, China; wangbingwei@boe.com.cn; 3School of Material Science and Engineering, University of Science and Technology of China, Hefei 230026, China; qbzhu15s@imr.ac.cn

**Keywords:** pressure sensor, carbon nanotube films, polystyrene microsphere, wide detection range

## Abstract

Flexible pressure sensors are important components of electronic skin and flexible wearable devices. Most existing piezoresistive flexible pressure sensors have obtained high sensitivities, however, they have relatively small pressure detection ranges. Here, we report flexible pressure sensors with a wide detection range using polydimethylsiloxane (PDMS) as the substrate, carbon nanotube films as the electrode material, and self-assembled polystyrene microsphere film as the microstructure layer. The obtained pressure sensor had a sandwich structure, and had a wide pressure detection range (from 4 kPa to 270 kPa), a sensitivity of 2.49 kPa^−1^, and a response time of tens of milliseconds. Two hundred load–unload cycles indicated that the device had good stability. In addition, the sensor was obtained by large-area fabrication with a low power consumption. This pressure sensor is expected to be widely used in applications such as electronic skin and flexible wearable devices.

## 1. Introduction

Electronic skin and flexible wearable devices are two important flexible electronic technologies. The development of electronic skin is motivated by growing interest in artificial intelligence, human-machine interfaces, and prosthetic skin [1,2,3]. Moreover, flexible wearable devices have promising potential for use in health monitoring and nursing applications [4,5]. Tactile sensing is one of the fundamental functions of skin, and therefore electronic skin will be developed based on flexible pressure sensors. Flexible pressure sensors can be categorized according to their working principles as piezoresistive, capacitance [6,7,8,9,10,11,12,13,14,15,16], piezoelectrics [17,18,19,20,21], or other working mechanisms, for example, triboelectric [22].

A pressure sensor is a detection device that reflects changes in pressure as changes in electrical signals. Piezoresistive flexible pressure sensors have been widely studied and applied due to their simple structures and working mechanisms, relatively simple fabrication processes, and excellent performance. Extensive research has been carried out to obtain flexible pressure sensors with excellent performances and high sensitivities. Micro-structured polydimethylsiloxane (PDMS) is commonly used as a substrate when combined with carbon nanotubes, graphene, or another conductive material as the electrode. This basic approach can be used to fabricate piezoresistive flexible pressure sensors [23,24,25,26,27]. For example, Ko et al. [28] used a silicon mold with periodic spherical holes to cast a mixture of carbon nanotubes and PDMS and obtained a pressure sensor with a maximum sensitivity of 15.1 kPa^−1^. Although microstructured PDMS films fabricated using microstructured silicon molds can be used to create highly sensitive devices, fabrication costs are relatively high due to the mold requirements, photolithography process, and other aspects of the fabrication process. Therefore, researchers have instead turned their attention towards nature [29,30]. For example, Zhang et al. [31] used natural leaf patterns as molds to obtain microstructured PDMS films, and the prepared sensors had high sensitivities (19.8 kPa^−1^). Elastic porous conductive materials are also widely used in piezoresistive flexible pressure sensors. Bao et al. [32] developed flexible pressure sensors based on an elastic microstructured conductive polymer with a high sensitivity (133.1 kPa^−1^).

The development of flexible pressure sensors with high sensitivities is undoubtedly the primary goal, and a significant amount of research has been carried out. However, pressure sensors which have only a high sensitivity are not ideal since detection range, power consumption, and other properties must also be considered. There are many reports of high-sensitivity piezoresistive flexible pressure sensors [33,34,35,36,37], but most piezoresistive devices have narrow detection ranges. A wide detection range is important for robotics and medical diagnostics [16]. In addition, these devices still consume power, even when no pressure is applied. Generally, the electrode of piezoresistive flexible pressure sensors is always switched on, even in the initial state, which produces a contact resistance. Power consumption under standby conditions is wasteful but can be reduced if the electrodes are designed to be non-conducting in the initial state. Gui et al. [38] reported a pressure sensor design using microporous SU8 photoresist as the isolation layer. When no pressure was applied, the SU8 separated the upper and lower electrode layers, and no current flowed, which reduced power consumption. 

Here, we report a piezoresistive flexible pressure sensor with a wide pressure detection range. The sensor uses a self-assembled single-layer polystyrene microsphere film as the microstructure layer, and a large-area carbon nanotube film as the flexible electrode. Its sensitivity reached 2.49 kPa^−1^, and its detection range was from 4 kPa to 270 kPa, which has not been previously achieved by most reported piezoresistive flexible pressure sensors. The response time was less than 100 ms, and the sensor was stable after 200 load–unload tests. The large-area preparation and low power consumption of this device have not been seen in most previously-reported piezoresistive devices. 

## 2. Materials and Methods

The piezoresistive flexible pressure sensor designed in this paper is shown in Figure 1. The sensor is a sandwich structure in which a single-layer self-assembled polystyrene (PS) microsphere film is located between two PDMS layers with a transferred carbon nanotube film. A top-down view of the self-assembled polystyrene microsphere monolayer film is shown in Figure 2a. There are obvious gaps between the microspheres, and the upper and lower carbon nanotube films contact each other through these gaps. The size of the gaps is related to the microsphere diameters, and the larger the diameter of the microsphere, the larger the gap between two spheres. As shown in Figure 2b, when no pressure was applied, the single-layer PS microsphere film separated the upper and lower carbon nanotube film layers, and the pressure sensor was in an off state, and no current flowed. When pressure was applied, the carbon nanotube film on the top layer contacted the bottom carbon nanotube film through a gap between the PS microsphere film, and the pressure sensor was turned on to generate a current. Larger pressures generated larger currents, and after the pressure was removed, the current was removed and the sensor was restored to its original state.

The carbon nanotube film was obtained via the continuous fabrication of meter-scale single-walled carbon nanotube film [39], reported in our previous work, which was collected on a filter membrane.

Self-assembled single-layer PS microsphere films were fabricated by first sonicating a suspension of PS microsphere powder in absolute ethanol for 10 min to ensure even mixing. The mass fractions of solutions containing PS microspheres with different diameters used in this paper were: 16.7 wt % (5 µm), 33.3 wt % (20 µm), and 33.3 wt % (5/20 µm), and the mass ratio of the two microspheres of the mixed solution was 1:5. Then a clean beaker was filled with an appropriate amount of DI water. The PS microsphere solution was slowly dropped into the deionized water along the inner wall of the beaker. Next, the prepared active agent solution (sodium dodecyl sulfate) was dropped into the beaker along the inner wall of the beaker to obtain a self-assembled PS microsphere film floating on the water surface. The surfactant reduced the surface tension of the water and more closely aligned the PS microspheres. Finally, the prepared PDMS substrate was inserted into the water at a 60° inclination angle, and then slowly pulled, and the PS microsphere film was transferred to the PDMS film and then allowed to dry.

The pressure sensor fabrication process shown in Figure 3a consisted of four steps. In the first step, an appropriately-sized cured PDMS film was placed on a rigid substrate. The PDMS film was obtained as follows: The PDMS mixture (monomer and curing agent: 10:1 weight ratio) was spin-coated using a spin coater. On the silicon wafer, cured PDMS (about 200 µm thick) was obtained by heating at 150 °C for 12 min. The wafer was treated with trichloro (1H, 1H, 2H, 2H-perfluorooctyl) silane to ensure that the cured PDMS could be torn off. In the second step, the carbon nanotube film was collected on a filter membrane. Then, a carbon tube film of a suitable size was placed in the middle of the PDMS film and gently pressed by finger. Then, isopropyl alcohol (IPA) was added dropwise to the surface, and after drying naturally. In this way, the carbon nanotube film was successfully transferred onto the PDMS substrate, and the isopropyl alcohol caused the carbon nanotube film to thicken and tighten, and bond with the substrate more firmly. In the third step, the PDMS substrate transferred with the carbon nanotube film was placed for use. Then, self-assembled single-layer PS microspheres were transferred to the PDMS substrate with the transferred carbon nanotube film by the pulling method and allowed to dry. The method of transferring the PS microspheres to CNT/PDMS is the same as the method of transferring to PDMS. This step also required the use of conductive silver glue to bond copper foil to the carbon tube film to extract the test electrode. In the fourth step, two PDMS substrates with transferred carbon nanotube films (one of which was transferred with a self-assembled single-layer PS microsphere film) were prepared, and two PDMS films were placed face-to-face to assemble a pressure sensor. Figure 3b is a photograph of the PS sensor.

The morphologies of the self-assembled PS microsphere monolayer films were characterized by an optical microscope (Nikon LV100ND, Nikon, Tokyo, Japan). The morphology of carbon nanotube films was characterized by scanning electron microscopy (SEM, ZEISS sigma 300, ZEISS, Jena, Germany). Sensor performance tests were performed using a mechanical performance testing machine (Instron 5943, C2MI, Bromont, QB, CANADA) to apply pressure to the sensor. A semiconductor device analyzer (Agilent B1500A, Agilent Technologies, Santa Clara, CA, USA) and connecting cables was used to apply a voltage (1 V) to the two electrodes and measure the corresponding electrical signal change.

## 3. Results and Discussion

Optical micrographs of self-assembled PS microsphere monolayer films are shown in Figure 4. Three flexible pressure sensors were designed using PS microspheres with different diameters. Figure 4a shows an optical microscopy image of a self-assembled single-layer of PS microspheres with diameters of 5 μm. The figure shows that the microspheres were arranged very tightly, with only a small gap between each sphere. Figure 4b shows an optical photo of self-assembled single-layer PS microspheres with a diameter of 20 μm. Compared with the 5 μm PS microsphere film, the gap between the spheres was significantly increased due to the larger diameter of the 20 μm PS microspheres. Figure 4c is an optical micrograph of a film obtained by mixing 5 µm and 20 µm PS microspheres in a mass ratio (1:5). The gaps in the mixed microsphere film were several tens of micrometers large, and different gap sizes were used to obtain three piezoresistive flexible pressure sensors with different gap structures.

A scanning electron micrograph of the carbon nanotube film is shown in Figure 5. It can be seen that long carbon nanotubes were densely and uniformly distributed throughout the film. The CNT film is a thin film network in which both metallic and semiconducting carbon nanotubes are mixed. When used as the electrode material of the pressure sensor, the metallic carbon nanotubes are primarily responsible for the functioning of the device.

The output current versus pressure curve of the flexible pressure sensor fabricated from three films with different microsphere diameters is shown in Figure 6. Figure 6a is the output curve of a pressure sensor fabricated with three different diameters (5 µm, 20 µm, and 5 and 20 µm) of PS microspheres. In the 5 µm device, as the pressure gradually increased from 0 to 530 kPa, the current remained unchanged at almost zero, and the device was in an open state. Obviously, such a device is not suitable for use as a sensor. An enlarged part of the curve for the 20 µm device is shown in Figure 6b, which shows that when the pressure was gradually increased to about 65 kPa, the device began to conduct, and a current was generated. It is useless because the required pressure to turn on the device is too large. In contrast, the device made by mixing PS microspheres with two different diameters showed good performance (Figure 6a). Combined with the PS microsphere photograph in Figure 4, it is clear that the 5 µm device has a small gap because of the small microsphere diameters, which prevented the device from being turned on under large pressures. With the diameter of the microspheres increased, the gap also increased, so the 20 µm device was able to operate. On the other hand, increasing the diameter also increased the height of the spacer between the carbon nanotube films, thereby increasing the difficulty of conduction between the upper and lower carbon nanotube films. Thus, the 20 µm device began conducting only at 65 kPa. In contrast, the device prepared by mixing the two microspheres had larger gaps, and the average height of the spacer was also moderate, and the device exhibited good sensing performance. The performance parameters of the devices fabricated with the hybrid microspheres are characterized in detail below.

Sensitivity is the most important performance metric for a sensor. For piezoresistive flexible pressure sensors, the sensitivity (*S*) can be calculated using Equation (1) [31]:
*S* = (Δ*I*/*I*_0_)/*P*(1)
where *S* represents the sensitivity of the pressure sensor, *I*_0_ represents the initial current value of the sensor when no pressure is applied, and Δ*I* is the change in the corresponding current when a pressure *P* is applied. This is used to calculate the sensitivity of most piezoresistive flexible pressure sensors. It should be noted that these sensors are also in a conducting state when no pressure is applied. However, the device reported in this paper is non-conducting under zero-pressure conditions, and the current can be regarded as zero, so the true sensitivity of the device cannot be obtained using Equation (1). Thus, this paper uses Equation (2) to calculate the sensitivity of the device [38]:
*S*(*P*) = *I*’(*P*)/*I*(*P*) = (d*I*/d*P*)/*I*(*P*)(2)
where *S*(*P*) represents the sensitivity of the pressure sensor, *I*(*P*) is the current at pressure *P*, and *I’*(*P*) is the derivative of current *I* at pressure *P*. The current-pressure curve of Figure 7 was analyzed, and the maximum sensitivity of the sensor calculated using Equation (2) was 2.49 kPa^−1^.

Most reported piezoresistive flexible pressure sensors have high sensitivities, however, they do not have sufficient detection ranges. Table 1 lists the detection ranges of pressure sensors reported in several representative studies, which clearly shows that these sensors have relatively small detection ranges. As shown in Figure 7, the pressure sensor reported in this paper has a wide measuring range (from 4 kPa to 270 kPa), and a linear detection range above 150 kPa, which has not been demonstrated in most piezoresistive flexible pressure sensors. The size of the gap between the microspheres is limited. When the applied pressure is increased to more than 300 kPa, the contact area of the upper and lower carbon nanotube films tends to be saturated, and the contact resistance does not change, so the current will be saturated. Within tolerance scope, it can be considered that the current does not change at an applied pressure of 400 kPa.

Response time and stability are also important performance indicators for sensors. As shown in Figure 8, the device in this work had a fast response time (<50 ms), as well as a good stability over 200 press–release pressure cycles. As shown in Figure 9, the device maintained a current of 40 µA under a pressure of 105 kPa. The error between the actual applied pressure and the set value of the universal testing machine used to apply pressure is the main reason why the current value fluctuated around 40 µA. The device is reliable enough to operate within tolerance scope. The self-assembled single-layer PS microsphere film can be used to achieve large-area fabrication, and the meter-scale carbon nanotube films can also be obtained via continuous fabrication. It is difficult to use large-area preparation techniques to obtain many reported pressure sensors due to limitations of the electrode materials or microstructured molds such as leaves. Obviously, flexible pressure sensors with large areas are more likely to be used for electronic skin that is similar to real skin. In addition, in the zero-pressure state, the device was in an open state and no current was generated, so the power consumption of the sensor is relatively low.

## 4. Conclusions

In this paper, piezoresistive flexible pressure sensors based on polystyrene microspheres were prepared. The sensors were made using PDMS as the substrate, a carbon nanotube film as an electrode, and polystyrene microsphere films as a microstructure layer to form a sandwich structure device. By using microspheres with different diameters, a device with a wide pressure detection range (from 4 kPa to 270 kPa) was obtained, which has not been achieved by most reported piezoresistive pressure sensors. The pressure sensor had a maximum sensitivity of 2.49 kPa^−1^ and a fast response time (<50 ms), as well as good stability after 200 load–unload cycles. In addition, the sensor also had a large area and low power consumption. The flexible pressure sensor reported in this paper demonstrated excellent performance and is expected to be suitable for use as electronic skin and in flexible wearable devices.

## Figures and Tables

**Figure 1 sensors-19-05194-f001:**
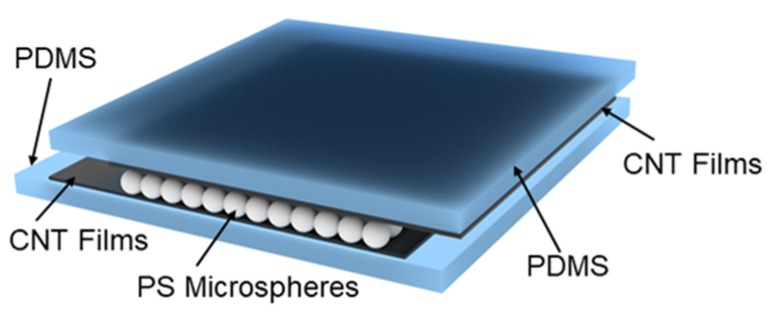
Schematic illustration of the pressure sensor. Polydimethylsiloxane (PDMS), polystyrene (PS) and carbon nanotube (CNT).

**Figure 2 sensors-19-05194-f002:**
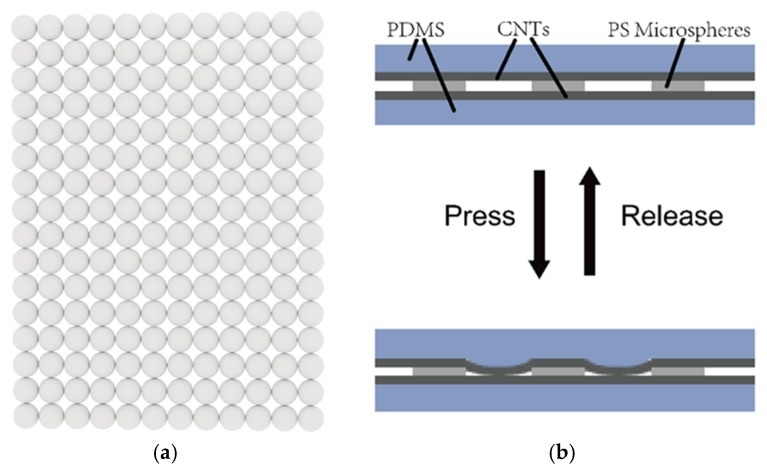
Schematic illustration of the device. (**a**) Top-down view of the PS microsphere films; (**b**) schematic illustration of the working principle.

**Figure 3 sensors-19-05194-f003:**
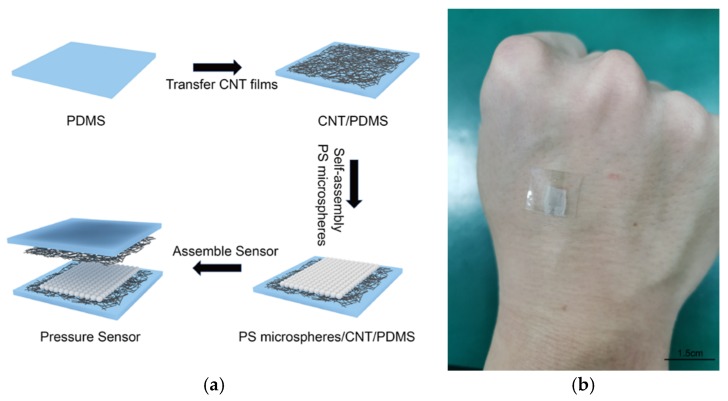
Fabrication procedure and photograph of the sensor. (**a**) Procedure for fabricating pressure sensor; (**b**) photograph of the PS Sensor.

**Figure 4 sensors-19-05194-f004:**
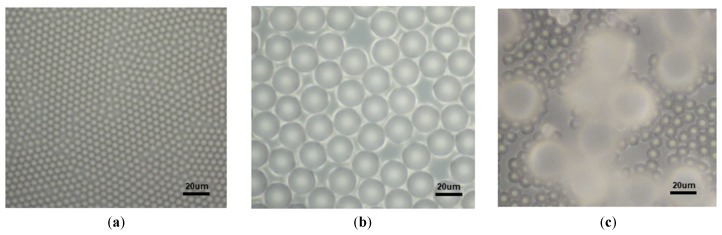
Optical photos of different PS microsphere films. (**a**) Diameter: 5 µm; (**b**) diameter: 20 µm; (**c**) diameter: 5 and 20 µm.

**Figure 5 sensors-19-05194-f005:**
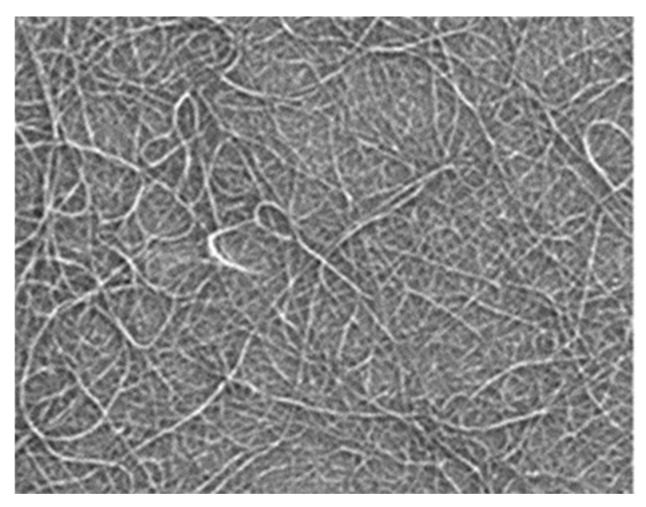
Scanning electron micrograph of CNT film.

**Figure 6 sensors-19-05194-f006:**
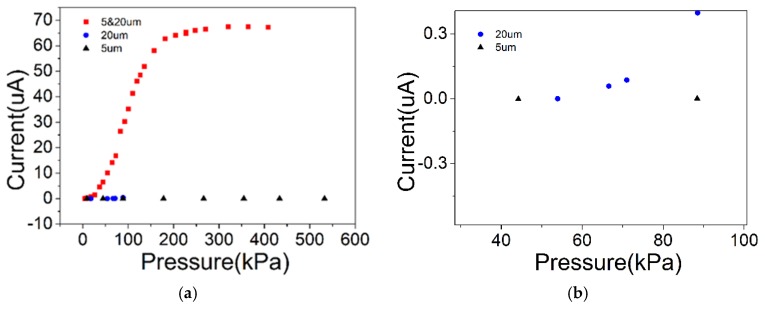
Current versus pressure curves of three kinds of sensors: (**a**) 5 µm, 20 µm, and 5 and 20 µm PS pressure sensors; (**b**) 5 µm and 20 µm PS pressure sensor (partial enlargement).

**Figure 7 sensors-19-05194-f007:**
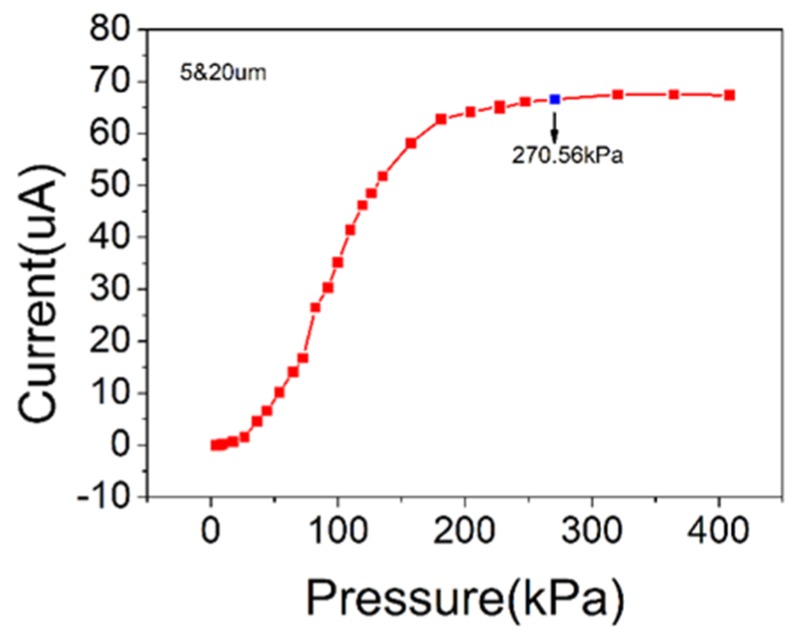
Current versus pressure curve.

**Figure 8 sensors-19-05194-f008:**
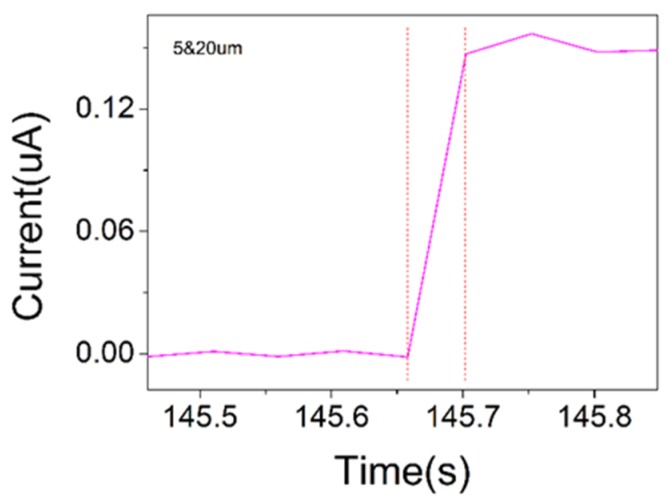
Response time of the PS sensor.

**Figure 9 sensors-19-05194-f009:**
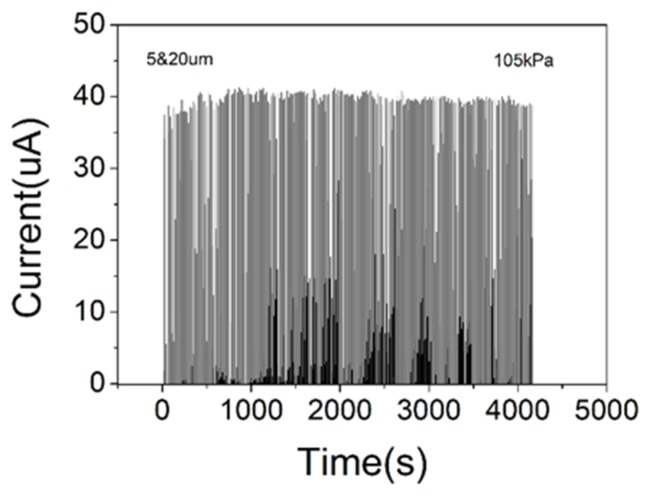
Durability test of the PS sensor.

**Table 1 sensors-19-05194-t001:** Comparison of device detection ranges.

Reference	[31]	[32]	[34]	[38]	[40]	This Work
Detection range (kPa)	6	10	1.2	15	700	270
Detection limit (Pa)	0.6	1	0.6	16	100,000	4000

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
