# Peer review of "Flexible Pressure Sensors with a Wide Detection Range Based on Self-Assembled Polystyrene Microspheres"

_sensors, 2019, doi:10.3390/s19235194_

Round 1
Reviewer 1 Report
In this paper, Yan et al. constructed a Flexible Pressure Sensor with Wide Detection Range. This research is interesting and novelty, and the results are well presented. I agree to accept it after minor modification was made.
(1) In Figure 6b, what do green dots mean?
(2) How do authors choose the mass ratio of the two microspheres? Did authors optimize it?
Author Response
We would like to thank the reviewer for the valuable and insightful comments. Some modifications and additions are made in this revised version in colors.
In this paper, Yan et al. constructed a Flexible Pressure Sensor with Wide Detection Range. This research is interesting and novelty, and the results are well presented. I agree to accept it after minor modification was made.
(1) In Figure 6b, what do green dots mean?
Answer: Figure 6 (b) illustrated that the 20μm device is turned on to generate a current at 65kPa. The figure has been modified. Please see the Figure 6 (b) on pages 6.
(2) How do authors choose the mass ratio of the two microspheres? Did authors optimize it?
Answer: We optimize the mass ratio of the two microspheres. On the one hand: the volume of the microsphere with a diameter of 20 μm is much larger than the 5μm-microsphere. Under the same quality, the former is far less than the number of the latter; On the other hand, we found that the minute quantity doping of large-diameter microspheres can obtain a microsphere film with a large gap. Finally, this Combining the two factors, after different mass ratio attempts, we obtain the final mass ratio.
On the whole, I revised the paper carefully according to the reviews. I wish now it can meet the requirements of Sensors. Many thanks to the reviewer again!

Reviewer 2 Report
This paper reports on the development of a highly sensitive and large range of detection pressure sensors. The sensing mechanism of the sensors is based on the increase in the contact area between two electrodes separated by a thin layer of polystyrene microspheres. The electrodes were formed using multiwall CNT which were transferred onto a PDMS substrate.
The fabrication of the sensors was relatively simple while the sensors exhibited good performance. The use of polystyrene microsphere as a separating layer is new. The review recommends this work in Sensors after a major revision.
Please address the following comments:
1. The description of the fabrication process is relatively long. There is some unnecessary information which can be removed. For instance, in line 96 “After the water surface was calm”: can be removed. Is it important to drop the active agent solution “along the inner wall”?
The fabrication of PDMS has been standardized, so just mention briefly (ratio 1:10, by spin-coating)
2. How can the authors confirm that only a single layer of the microsphere was attached on to PDMS but not multiple layers?
3. How the PS microspheres were transferred onto CNT/PDMS? Did the author use another stamping step here? Is the van de Wall force between CNT and PS strong enough to hold this layer upon removing the hosting PDMS?
4. Some sentences do not have any meaning. Please check the typo and grammar errors carefully. For instance: in line 162 “Although this device sensing properties”
5. In figure 6(b), what are the blue dots? The red squares seem not necessary, as they are out of range.
6. Please explain why the current saturated at applied pressures above 300kPa. From Figure 4, the current seemed to decrease at an applied pressure of 400kPa.
7. The authors compared the detection range of the developed sensors with the other ones reported in the literature in Table 1. However, in this table the authors only listed the upper limitation. What about the minimum detectable pressure level of the other sensors? The reviewer expects that although these sensors only detect much lower range of pressure, they may have much smaller turn-on pressure level.
8. The concept of the pressure sensors using contacting mode has been reported in the literature. For instance, Fastier-Wooller et al. developed pressure sensors using paper sandwiched between two conductive layers (https://www.mdpi.com/1424-8220/18/10/3300/htm). This paper also employs natural structures and low-cost materials, while offering very large sensitivity range (from 150 kPa up to 1000kPa). The authors should compare their results to the previous reports in the manuscript.

Round 2
Reviewer 2 Report
The authors have addressed all of my comments. Therefore, I would like to recommend this work for publication in Sensors as is.
Further English proofread may be required to avoid any typo, and grammatical errors in the manuscript.